# Effects of Water Depth and Phosphorus Availability on Nitrogen Removal in Agricultural Wetlands

**Xiaojun Song [1,2], Per Magnus Ehde [1] and Stefan E. B. Weisner [1,]**\* 

[1] Rydberg Laboratory for Applied Sciences, Halmstad University, 301 18 Halmstad, Sweden; xiaoso17@student.hh.se (X.S.); per_magnus.ehde@hh.se (P.M.E.)
[2] School of Environmental Science and Engineering, Suzhou University of Science and Technology, Suzhou 215000, China
\* Correspondence: stefan.weisner@hh.se

**Abstract:** Excess nitrogen (N) from agricultural runoff is a cause of pollution in aquatic ecosystems. Created free water surface (FWS) wetlands can be used as buffering systems to lower the impacts of nutrients from agricultural runoff. The purpose of this paper was to evaluate critical factors for N removal in FWS wetlands receiving high nitrate ($NO_3^-$) loads from agriculture. The study was performed in 12 experimental FWS wetlands in southern Sweden, receiving drainage water from an agricultural field area. The effects of water depth (mean depth of 0.4 m and 0.6 m, respectively) and phosphorus (P) availability (with or without additional P load) were investigated from July to October. The experiment was performed in a two-way design, with three wetlands of each combination of depth and P availability. The effects of P availability on the removal of $NO_3^-$ and total N were strongly significant, with higher absolute N removal rates per wetland area (g m$^{-2}$ day$^{-1}$) as well as temperature-adjusted first-order area-based removal rate coefficients ($K_{at}$) in wetlands with external P addition compared to wetlands with no addition. Further, higher N removal in deep compared to shallow wetlands was indicated by statistically significant differences in $K_{at}$. The results show that low P availability may limit N removal in wetlands receiving agricultural drainage water. Furthermore, the results support that not only wetland area but also wetland volume may be important for N removal. The results have implications for the planning, location, and design of created wetlands in agricultural areas.

**Keywords:** free water surface wetlands; agricultural runoff; nitrogen removal; water depth; phosphorus

## 1. Introduction

The use of nitrogen (N) fertilizers in agriculture has increased dramatically during the last century and is expected to continue to increase in most future scenarios [1]. Agricultural runoff, which is rich in nutrients, is a source of diffuse pollution and contributes to eutrophication [2,3]. In recent decades, wetlands have been created for the remediation of runoff from agriculture and are now used to reduce N transports at watershed scales [4].

The application of agricultural treatment wetlands has been reported in several countries worldwide. For example, from 1996 to 2008, over 2400 wetlands were created in Sweden with support from different national programs, with the aim of remediating runoff from agriculture and decreasing N transport to the eutrophicated Baltic Sea [5]. In the US, the ongoing restoration and creation of wetlands along the Missouri River is part of a strategy to decrease N loads to the Mississippi River and eventually, to the Gulf of Mexico [6]. In Ireland, the Integrated Constructed Wetland concept

has been used to address the pressure from different EU directives to improve water quality at the landscape scale since 2007 [7].

Free water surface flow (FWS) wetlands, where the water surface is exposed to the atmosphere, are prevalent among created agricultural wetlands [8,9]. They may contain areas of e.g., open water, floating-leaved vegetation, and emergent plants either due to planting or as a consequence of wetland design [8]. The creation of this kind of wetland is a cost-efficient treatment measure that can be integrated into agricultural landscapes to deal with eutrophication [10,11].

The mechanisms of N removal in FWS wetlands are manifold and include e.g., volatilization, microbial assimilation, plant uptake, and nitrification/denitrification [12]. Denitrification typically accounts for most of the nitrate ($NO_3^-$) removal in constructed wetlands and is an important N removal mechanism, especially in agricultural wetlands, as the N in drainage water from agricultural fields is largely in the form of $NO_3^-$ [13–17]. This microbial process converts $NO_3^-$ into gaseous N, thereby providing a removal pathway for N from the aquatic environment [18–20].

Water depth is a design parameter that can affect the treatment performance of FWS wetlands by affecting the hydraulic performance and vegetation abundance [21–24]. Holland et al. [25] showed, in a tracer study, that deeper water can increase the residence time for water in wetlands. Wu et al. [26] suggested that there will be more time for contaminant removal at longer hydraulic residence times with deeper water. Arheimer and Wittgren [27] suggested that created wetlands should have a residence time of at least 2 days to be considered to contribute to the reduction of N loads on a catchment scale.

On the other hand, some studies have suggested that shallow water in FWS wetlands results in a higher N removal efficiency. Liu et al. [28] studied the influence of water depth with a tracer test and model study. This study showed that increased water depth may cause dispersion of the pollutants, lower the concentration gradient, and reduce the reaction rate. Sanchez-Ramos et al. [29] found that shallow water allows a greater development of bacterial communities, which has an impact on the biochemical reactions triggering contaminant transformation and degradation.

Although N removal processes in created wetlands are largely physical and microbial, the role of macrophytes is significant [30]. Macrophytic vegetation assimilates nutrients in wetlands from the water as well as the litter layer/sediment [31,32]. As a result of this, nutrients will be incorporated into plant biomass [33]. Removal of biomass through harvesting is necessary to realize the full nutrient removal potential of macrophyte uptake, as the harvesting prevents the release of nutrients back into overlying water from plant residue decomposition [34]. However, plants in wetlands provide organic matter and suitable attachment surfaces to denitrifying bacteria, and they promote the development of anaerobic conditions through litter accumulation and decomposition, which all favor denitrification [14,35]. The importance of the effects of vegetation on N removal is evident from studies showing the positive influence of highly productive emergent vegetation on N removal in wetlands [36,37].

Water depth determines the cover and structure of macrophytes and the microbial growth on surfaces of vegetation in wetlands [17,38]. Specifically, the development of highly productive emergent vegetation is promoted by shallow water [39,40]. Weisner et al. [14] suggested that a rational distribution of macrophytes with different water depths in wetlands may promote denitrification processes and decrease the degree of short-circuiting. Furthermore, wetland vegetation may lead to the differentiation of water residence times [41], promoting N removal. However, Bodin et al. [42] indicated that dense vegetation could reduce the effective volume of wetlands, mainly because of the volume the plants occupy. Keefe et al. [43] indicated that during the growing season, new growth of emergent vegetation enhanced hydraulic performance, while at the end of the growing season, senescing vegetation created short-circuiting, which can decrease hydraulic performance. These divergent conclusions about the effects of vegetation complicate our understanding of the effects of water depth on N removal in FWS wetlands.

Phosphorus (P) is a key limiting nutrient that can influence the growth of macrophytes [44,45]. An increased growth of macrophytes, due to a higher P availability, may therefore result in an increased uptake of N by transforming into biomass, or it may indirectly promote denitrification by providing organic carbon or suitable surfaces for bacteria. Furthermore, there are a few studies indicating that denitrification rate is directly correlated to P availability. White and Reddy [46] found that potential denitrification rates were positively correlated to the total P of the wetland soil as a result of P enrichment in soils affecting microbial activities. Li et al. [47] indicated that denitrification rate in FWS constructed wetland sediment/soil was positively correlated with the concentrations of the total P in treated river water. Kim et al. [48] also came to similar conclusions based on the finding that sediment denitrification rates were higher in P-enriched sites than in unimpacted sites in the Everglades wetlands. However, studies on relationships between P availability and N removal in agricultural FWS wetlands are still very limited.

The aim of this paper was to evaluate the effects of water depth and P availability on N removal in FWS wetlands receiving N largely as $NO_3^-$. This knowledge may aid the design and management of agricultural wetlands. The hypotheses of this study are: (1) shallow water will result in a higher N removal efficiency in FWS wetlands, and (2) increased P availability may increase the efficiency of N removal.

## 2. Materials and Methods

The study was performed in an experimental wetland area with eighteen FWS wetlands (Figure 1) located near Halmstad in the south west of Sweden (56°43′45″ N, 12°43′33″ E). These wetlands were constructed in 2002, consisting of 18 similar rectangular wetland basins [36,37,42]. Each of the wetland basins had a flat bottom area of 12 m$^2$ (1.6 m × 7.6 m) and a ground surface area of 40 m$^2$ (4 m × 10 m), with a side slope of 1:1. Plant cuttings (emergent and submerged macrophytes) were introduced into some wetlands in 2003, while some of the wetlands were kept unplanted to achieve freely developing vegetation. This study was performed in 2017 when vegetation in all wetlands was well established. Twelve out of 18 wetlands, with similar vegetation dominated by *Phragmites australis* and *Typha latifolia*, were chosen to be included in this study.

The incoming water to the experimental wetlands was drainage water from agricultural fields. Therefore, the inflow water to the wetlands had relatively high N concentrations, almost entirely composed of $NO_3^-$ (average inflow $NO_3$–N and Tot-N concentrations during the experiment were 7.7 and 8.1 mg L$^{-1}$, respectively), but low P concentrations (Tot-P was less than 0.01 mg L$^{-1}$ in inlet water to the wetlands). Incoming water to the experimental area was first distributed to three tanks, then distributed through ground pipes to six wetlands each (Figure 1). The water flow could be regulated by a valve on the inlet pipe of each wetland. Furthermore, each wetland had an outlet pipe that could be adjusted vertically to regulate water depth. Prior to the experiment, the water level had been set to a mean water depth of 0.6 m since 2006. To achieve different water depths, 6 wetlands out of 12 were randomly selected to be controlled as "shallow", which were adjusted to a mean depth of 0.4 m in June 2015 by lowering the outlet pipe. The remaining wetlands were denoted as "deep". Three of the 6 wetlands of each water depth treatment were randomly selected to be wetlands with an additional P load.

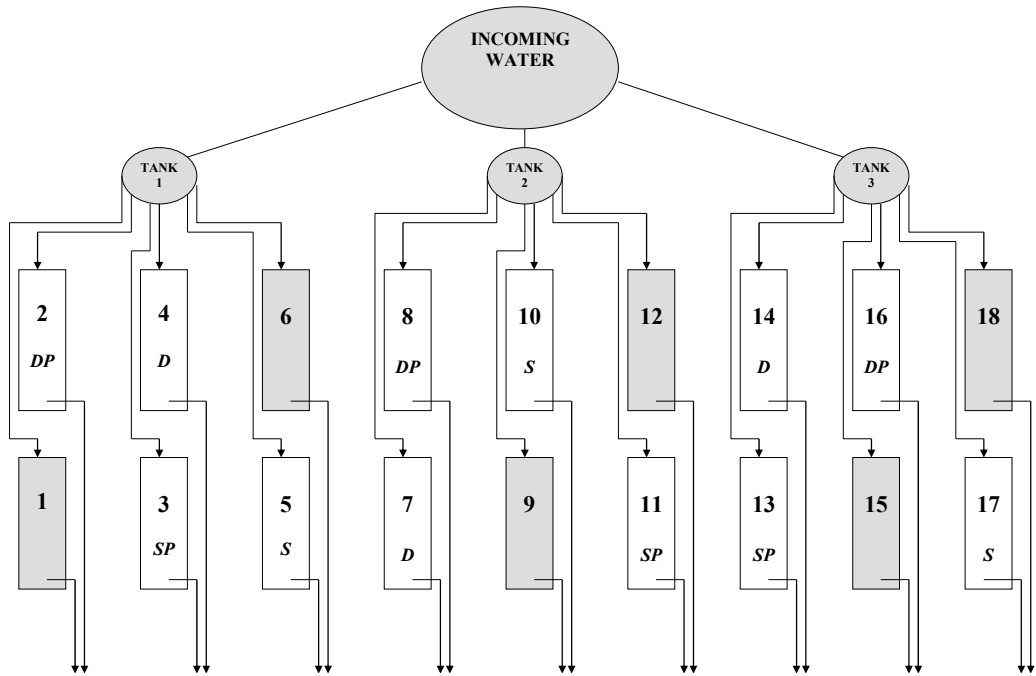

**Figure 1.** Position of the 18 wetland basins (10 × 4 m) within the experimental area (7632 m). DP: deep wetland with P addition; D: deep without P addition; SP: shallow with P addition; S: shallow without P addition; grey wetlands: not included in this study.

The average water surface area was smaller for shallow wetlands than for deep wetlands (18.9 and 26.8 m$^2$, respectively, as an effect of sloping sides and lowering the water level). The water flow was adjusted regularly to obtain a similar areal hydraulic load (0.1 m day$^{-1}$) to all the wetlands. However, the average areal hydraulic load during the experiment was slightly higher in shallow wetlands than in deep wetlands (0.119 and 0.106 m day$^{-1}$, respectively). Phosphorus additions were performed on 7 occasions from July to October, one week before water sampling occasions. To obtain a similar additional areal P-load to shallow as well as deep wetlands, 70 or 48 g $KH_2PO_4$ (corresponding to 16 and 11 g P) was dissolved in 1 L glass bottles and sprinkled over the water surface at the inlet section of three deep and three shallow wetlands, respectively. This resulted in a P-load corresponding to an average inlet P concentration of 0.4 mg L$^{-1}$ from July to October in these 6 wetlands.

Water samples to estimate N removal were collected on 7 occasions from 6 July 2017 to 17 October 2017 with a frequency of every second or third week. On each occasion, incoming water was collected from each of the three inlet tanks. On the same occasions, outflow water was collected and water flow and water temperature were measured at the outlet of each wetland. Precipitation on wetland surfaces may lead to a dilution effect resulting in an overestimation of N removal. Therefore, water sampling within 3 days of precipitation was avoided to minimize the influence of the precipitation. Samples were analyzed for concentrations of $NO_3$–N and total-N with flow injection analysis using standard photometric methods on a FIAstar 5000 Analyzer (FIAstar, FOSS, Munkedal, Sweden). Tot-N was analyzed after the digestion of the soluble reactive phases ($NO_3^-$ and $NO_2^-$) by exposure to potassium peroxide. The average water temperature during the experiment was similar in shallow and deep wetlands (13.6 and 13.5 °C, respectively) and was highest in August (14.4 °C for both treatments) and lowest in October (12.0 and 11.7 °C, respectively). Precipitation was measured at the experimental location and was 46, 90, 96, and 48 mm during July, August, September, and October (until the last sampling on 17 October), respectively. The pH was measured in the wetland outlets in July and varied among wetlands between 7.28 and 7.64.

The absolute removal (g m$^{-2}$ day$^{-1}$) and a temperature-adjusted first-order area-based removal rate coefficient ($K_{at}$, m day$^{-1}$) that has previously been proved applicable to created wetlands [8,49]

were calculated for $NO_3$–N and Tot-N for each wetland and sampling occasion. This was based on the inflow and outflow of N concentrations, the measured water flow, and the temperature, at each occasion. The effects of precipitation and evapotranspiration were assumed to be small in comparison to the hydraulic load to the wetlands and to have the same effect in all the wetlands, therefore not affecting comparisons between treatments. Inflow N-concentrations were measured on the water samples from the three inlet tanks. The median of these 3 values was used to represent the inflow concentration to all the wetlands at each occasion to avoid errors caused by extreme occasional values.

When calculating temperature-adjusted first-order area-based removal coefficients, the tank-in-series model approach was used in Equation (1), combined with a modified Arrenhius temperature dependency in Equation (2), to adjust the coefficient to represent what the removal rate would have been at a "standard" temperature of 20 °C [49]:

$$C_{out} = C_{in}(1 + \frac{(\frac{A}{Q})K_{at}}{N})^{-N} , \tag{1}$$

$$K_{at} = K_a \times \theta^{(T-20)}, \tag{2}$$

where $C_{out}$ is the N concentration in the outflow (mg $L^{-1}$), $C_{in}$ is the N concentration in the inflow (mg $L^{-1}$), $Q$ is water flow ($m^3$ $day^{-1}$), $A$ is the wetland surface area ($m^2$), and $T$ is the water temperature (°C). $N$ is the number of tanks in series in the model, expressing the degree of plug flow, which was assumed to be equal in all wetlands. $N$ was set to 2 according to a previous hydraulic study in the experimental wetlands [42]. The temperature coefficient $\theta$ was set to 1.088, as previously suggested for created wetlands [49].

Differences in N-removal among wetland treatments were analyzed with repeated-measures analysis of variance (ANOVA), with the fixed factors water depth and P addition. Effects were accepted as statistically significant at $p < 0.05$.

## 3. Results

The statistical analysis (Table 1) reveals strongly significant effects of P addition on absolute removal as well as removal rate coefficients ($K_{at}$) for $NO_3^-$ and Tot-N. There was no statistically significant difference between deep and shallow wetlands in the absolute removal of $NO_3^-$ and Tot-N. However, $K_{at}$ differed significantly between depth treatments for $NO_3^-$ as well as Tot-N. There was no significant interaction between the two treatments (P addition and water depth), indicating that the effect of one treatment was not affected by the other treatment.

**Table 1.** Between-subject effects (P-values) of wetland treatments on $NO_3$–N and Tot-N removal (absolute removal and temperature-adjusted removal rate coefficients, $K_{at}$) according to repeated-measures analysis of variance (ANOVA) based on 7 repeated measures (sampling occasions). The treatments were P-addition (Phosphorus) and water depth (Depth). Interaction effect is denoted as Phosphorus * Depth.

| *Absolute Removal* | NO$_3$-N | Tot-N |
| --- | --- | --- |
| Phosphorus | 0.001 | 0.001 |
| Depth | 0.057 | 0.063 |
| Phosphorus * Depth | 0.578 | 0.724 |
| *Removal Rate Coefficient* | NO$_3$-N | Tot-N |
| Phosphorus | 0.000 | 0.001 |
| Depth | 0.006 | 0.001 |
| Phosphorus * Depth | 0.102 | 0.481 |

The absolute removal of Tot-N followed a seasonal pattern with higher removal rates in July to September (Figure 2). The absolute removal of Tot-N was consistently higher throughout the 4 months

of the study (when comparing within each month) in wetlands with P addition compared to wetlands with the same water depth but without P addition. Deep wetlands generally, but not consistently, exhibited higher average absolute Tot-N removal rates than shallow wetlands with the same P treatment, when comparing within each month. As mentioned above (Table 1), the effect of P addition was strongly statistically significant, but the effect of water depth was not statistically significant.

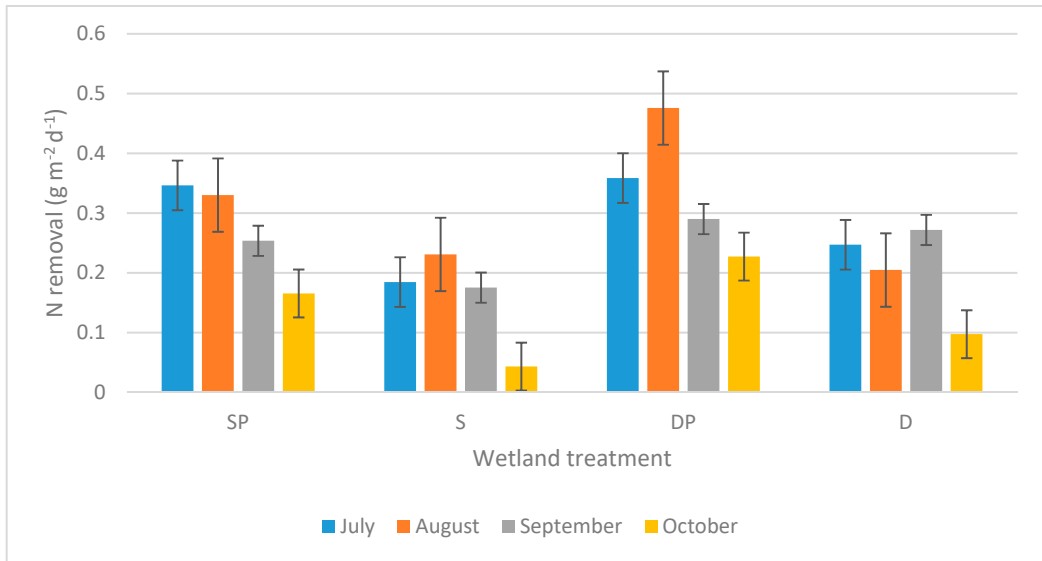

**Figure 2.** Average absolute Tot-N removal for different wetland treatments during different months. S = shallow wetlands, D = deep wetlands, P = with P addition. The error bars show standard errors.

Additionally, the temperature-adjusted removal rate coefficient ($K_{at}$) for Tot-N was consistently higher for each of the 4 months of the study in wetlands with P addition, when comparing to wetlands with the same water depth but without P addition (Figure 3). Furthermore, deep wetlands, consistently throughout the study, exhibited clearly higher simultaneous $K_{at}$-values than shallow wetlands with the same P treatment. This matches the strongly statistically significant effects for both treatments according to repeated-measures ANOVAs (Table 1).

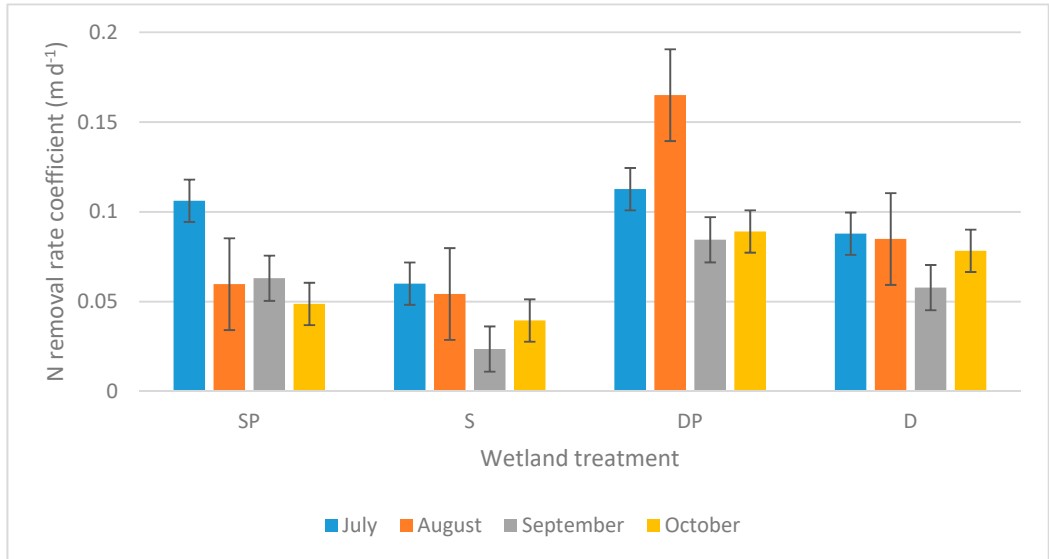

**Figure 3.** Average first-order area-based removal rate coefficients ($K_{at}$, m day$^{-1}$) for Tot-N for different wetland treatments during different months. Coefficients are temperature adjusted to 20 °C. S = shallow wetlands, D = deep wetlands, P = with P addition. Error bars show standard errors.

## 4. Discussion

In this study, water depth did not have a statistically significant effect on absolute N removal per wetland area. However, the effect of depth treatment on the removal rate coefficient ($K_{at}$) for Tot-N was strongly statistically significant with higher values in deep wetlands, showing that first-order area-based rate coefficients were essential in this study to reveal differences in N removal between treatments. The reason can, in this case, be that $K_{at}$ is adjusted for differences between individual wetlands in water temperature and hydraulic load ($K_{at}$ is also adjusted for the inlet concentration, but this concentration was the same for all wetlands in this study). Therefore, this study suggests that deep wetlands have a higher N removal capacity per wetland area compared to shallow wetlands, which opposes our hypothesis (1) that shallow water will result in a higher N removal efficiency in FWS wetlands. The effects of P availability on the removal of $NO_3^-$ and total N were clearly strong, with significantly higher absolute N removal per wetland area as well as $K_{at}$-values in wetlands with external P addition compared to wetlands with no addition. This supports our hypothesis (2) that increased P availability will increase the efficiency of N removal.

The effects of water depth on N removal can be explained by the residence time. The wetlands in this study had similar hydraulic loads per wetland area. Thus, the water residence time was longer in the deeper wetlands. The treatment efficiency of a wetland is, to a great degree, governed by a longer residence time as it gives more time for reactions to occur [26]. We suggest that longer residence times in the deep wetlands in this short-term study promoted N removal. However, it is likely that shallow wetlands will successively develop more vegetation than deeper wetlands [39,40], and therefore, that positive effects on the N removal of vegetation in shallow wetlands will become more important in the long run.

The effects of P addition confirm previous studies that have reported that denitrification rate is correlated to the P availability, as a higher P availability can positively affect microbial denitrification [44,46]. Furthermore, increased plant production due to P additions also could have contributed to N removal through increased plant uptake of N as well as by providing organic carbon or surface areas to denitrifying bacteria [14,35]. The effect of P addition has implications for the creation of wetlands with the purpose to remove N from agricultural drainage water, as concentrations of P, especially directly bioavailable $PO_4$-P, can be very low in drainage water from agricultural fields in spite of high N-concentrations [50,51]. Therefore, the results of this study implicate that N removal in wetlands receiving drainage water from agricultural fields can be limited by P deficiency.

This study emphasizes the importance of water depth and external P load on N removal in FWS created wetlands. The results suggest that not only wetland area but also wetland volume may be important for N removal. However, a higher plant production in shallow wetlands [17,38–40] may promote N removal [36,37], which could not be tested in this short-term study. The results suggest that a low availability of P may, in some situations, limit N removal in wetlands receiving drainage water from agricultural fields.

One practical implication of the results in this study is that a large wetland area as well as a large wetland volume should be strived for when planning wetland creation in agricultural areas for N removal. It is most likely beneficial if a larger wetland volume is achieved by increasing the wetland area rather than the depth, since a vegetation development promoting N removal can be expected in shallow wetlands. Further, the costs of increasing water depth must be compared to the costs of increasing the wetland area. Another practical implication is that N removal in wetlands is affected by P availability, which may differ considerably among wetlands in agricultural areas due to, e.g., soil conditions in the watershed as well as if the wetland is fed by stream water or drainage water [10,50,51]. Therefore, the planning of wetland creation for N removal may benefit from including P availability, e.g., by placing wetlands where streams can supply the wetland with P-rich water. Further research is needed to develop models including the impact of P availability on N removal in wetlands. Such models could be used for choosing the best locations for wetland creation to obtain a higher N removal per wetland area or cost.

## 5. Conclusions

Two critical factors (water depth and external P load) for N removal in FWS wetlands that had high levels of N but relatively low P concentrations were investigated in this study. The results show that a low availability of P is likely to limit N removal in agricultural wetlands. Furthermore, deeper wetlands with longer residence times promote N removal in the short term. However, a stronger vegetation development in shallow wetlands may promote N removal in the long run more than could be shown in this study.

**Author Contributions:** X.S. did a literature review for background information, statistical analysis of the data, and wrote a Master's thesis which constitute the basis for this paper; P.M.E. was responsible for the field sampling and chemical analyses; S.E.B.W. conceived and designed the experiment and completed the paper with contributions from the co-authors.

**Funding:** This research received no external funding.

**Acknowledgments:** This study was performed at the Wetland Research Centre at Halmstad University. It was supported by the Rydberg Laboratory for Applied Sciences (RLAS) at Halmstad University. We wish to thank Delila Hasovic for field and lab assistance.

**Conflicts of Interest:** The authors declare no conflict of interest.

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
