# Peer review of "Effects of Water Depth and Phosphorus Availability on Nitrogen Removal in Agricultural Wetlands"

_water, doi:10.3390/w11122626_

Round 1

Reviewer 1 Report

The manuscript contributes to enriching our knowledge about the functioning of constructed wetlands purifying water polluted by agrochemicals. The study provides an experimental explanation of two controls responsible for nitrogen removal, which are water depth and P availability. The results may be useful for the design and management of treatment wetlands in agricultural landscapes. 

The presented manuscript covers the current environmental problem and delivers interesting and inspiring results. Obviously, the Authors put a lot of effort into the experimental work and also in writing this manuscript, which I would like to acknowledge here. I also want to recognize the precise description of the experiment, which is somewhat rare in literature.

The paper, for the most part, is clearly written and would be a good contribution to the Water. I do have, however, few comments which I outline below for the Authors to consider as part of further improving this piece of work before publication. 

Line 28 - nitrogen fertilization was always crucial for agriculture 

Line 29 - should be “is expected” 

The Introduction section is lengthy, as is contains a relatively extensive review of the literature on the subject addressed in the article. However, it seems that some of the material could be moved to the Discussion section, which is a bit poor. 

Line 116 - replace “plant communities” with “plant cuttings” or similar 

Line 121 - the phrase “groundwater which was fed by drainage water from agricultural fields” is unclear 

The manuscript confirmed the well known and quite obvious relationship between retention time and removal efficiency of N. It proved that low availability of P was likely to limit N removal in agricultural wetlands. How can this knowledge be used in the practice of designing and managing of constructed marshes treating groundwater from crop fields, which was pointed as an objective of the study? The controlling (or designing) of retention time is relatively easy to implement, but how to regulate the content of phosphorus in inflowing water? Additional fertilization of wetland with P from external sources is somewhat out of the question since phosphorus is a precious and shrinking resource. I would like to see the authors' comments on this issue.

Author Response

Dear reviewer,

Thanks for valuable comments! Here is our response (in capital letters):

Line 28 - nitrogen fertilization was always crucial for agriculture  SENTENCE MODIFIED 

Line 29 - should be “is expected”   CORRECTED

The Introduction section is lengthy, as is contains a relatively extensive review of the literature on the subject addressed in the article. However, it seems that some of the material could be moved to the Discussion section, which is a bit poor.  INTRODUCTION SHORTENED, REFERENCES ADDED TO DISCUSSION, PARAGRAPH ABOUT PRACTICAL IMPLICATIONS ADDED TO DISCUSSION

Line 116 - replace “plant communities” with “plant cuttings” or similar  DONE

Line 121 - the phrase “groundwater which was fed by drainage water from agricultural fields” is unclear  SIMPLIFIED TO MAKE CLEARER

How can this knowledge be used in the practice of designing and managing of constructed marshes treating groundwater from crop fields, which was pointed as an objective of the study? The controlling (or designing) of retention time is relatively easy to implement, but how to regulate the content of phosphorus in inflowing water?  PARAGRAPH ABOUT PRACTICAL IMPLICATIONS AND CONSIDERING THESE ASPECTS ADDED TO DISCUSSION

THANKS AGAIN

Reviewer 2 Report

The manuscript is in general well-written and easy to understand. The novelty is also well explained.

However, I have some concerns about your calculations and interpretations.

Lines 46-53: it is too much simplified. See some following comments:

Line 46: is anammox relevant for N-removal in FWS wetlands? And DNRA?

Ammonification does not remove N; in fact, it results in total N (NH4+) increase

What about microbial assimilation?

Line 50-52: First of all, what about autotrophic denitrification that doesn’t need any organic carbon addition. Have rates of this process been measured in FWS wetlands? Second, doesn’t a part of removed NO3- escaped to the atmosphere also as N2O?

Line 59: please define ‘hydraulic efficiency’

Line 71: if you write ‘on the one hand’ there should be also ‘on the other hand’

Line 134: why were areas different in size? According to the line 115 they are supposed to be the same in size.

Line 148-149: were outflow samples also collected in triplicates? It is unclear

Was pH measured? It could be of importance for e.g. ammonia volatilization.

Line 163: why did you use temperature of 20 C? Was it the average temperature during the 4 months of the experiment?

Was water loss via (evapo)transpiration taken into account? In free water surface wetlands that could be especially significant, and especially at 20 C. This could affect all your calculations. Also, precipitation (in mm of rainfall) had to be taken into account. That is why it is best to express removal efficiencies in loads.

Line 174-176: how do you explain a lack of statistically significant difference between depth for the absolute removal but presence for removal rate coefficient?

Line 183-188, 192-196: were the differences statistically significant?

Line 208-210: could higher N removal in deep wetlands compared to shallow wetlands be related to higher evaporation rate from shallow wetlands and therefore, N would get more concentrated within them? Did you measure in situ temperatures in wetlands? that could be higher in shallow wetlands. And in that case you would measure higher concentration in the outflow of shallow wetlands but effectively the removal efficiency per volume is higher than in deep wetlands?

Line 219-222: do you have a reference for this?

Author Response

Dear reviewer,

Thanks for valuable comments! Here is our response (in capital letters):

Lines 46-53: it is too much simplified.   BASED ON OTHER REVEIWERS COMMENTS WE NOW TRY TO MAKE THE INTRODUCTION SHORTER AND TO EXCLUDE DETAILS THAT ARE NOT IMPORTANT FOR THIS STUDY AND WETLANDS RECEIVING NITROGEN ALMOST ENTIRELY AS NITRATE

Line 46: is anammox relevant for N-removal in FWS wetlands? And DNRA?   WE REFORMULATED TO MAKE CLEARER THAT WE JUST GIVE SOME EXAMPLES OF NITROGEN REMOVAL PROCESSES IN THIS SENTENCE 

Ammonification does not remove N; in fact, it results in total N (NH4+) increase   REMOVED

What about microbial assimilation?  ADDED, REPLACING "MATRIX ADSORPTION"

Line 50-52: First of all, what about autotrophic denitrification that doesn’t need any organic carbon addition. Have rates of this process been measured in FWS wetlands? Second, doesn’t a part of removed NO3- escaped to the atmosphere also as N2O?  WE REFORMULATED SO WE ARE NOT IMPLYING THAT DENITRIFICATION ALWAYS NEEDS ORGANIC CARBON, AND ALSO SO WE DO NOT SPECIFICALLY MENTION N2 AS THE GASEOUS N

Line 59: please define ‘hydraulic efficiency’ UNCLEAR TERM SO WE REMOVED IT

Line 71: if you write ‘on the one hand’ there should be also ‘on the other hand’   REFORMULATED

Line 134: why were areas different in size? According to the line 115 they are supposed to be the same in size.  THIS IS AN EFFECT OF SLOPING SIDES AND DIFFERENT WATER LEVELS. WE NOW TRY TO EXPLAIN THIS

Line 148-149: were outflow samples also collected in triplicates? It is unclear   CLARIFIED SO IT SHOULD BE CLEARER THAT INFLOW WATER WAS SAMPLED IN THREEE INLET TANKS, NOT IN EACH WETLAND, BUT OUTFLOW WAS SAMPLED IN EACH WETLAND

Was pH measured? It could be of importance for e.g. ammonia volatilization. PH MEASUREMENTS ADDED

Line 163: why did you use temperature of 20 C? Was it the average temperature during the 4 months of the experiment?  IT IS TO OBTAIN A TEMPERATURE INDEPENDENT REMOVAL COEFFICIENT - WE TRY TO MAKE THIIS CLEARER NOW AND HAVE ALSO INCLUDED TEMPERATURE IN THE METHODS

Was water loss via (evapo)transpiration taken into account? In free water surface wetlands that could be especially significant, and especially at 20 C. This could affect all your calculations. Also, precipitation (in mm of rainfall) had to be taken into account. That is why it is best to express removal efficiencies in loads. WETLANDS WERE COLDER AND WE NOW MENTION PRECIPITATION AND EVAPOTRANSPIRATION AND HOW IT WAS CONSIDERED

Line 174-176: how do you explain a lack of statistically significant difference between depth for the absolute removal but presence for removal rate coefficient?  WE NOW MENTION THAT HYDRAULIC LOADS PER AREA WERE SLIGHLY DIFFERENT BETWEEN DEPTHS WHICH MAY HAVE AFFECTED THE RESULTS FOR ABSOLUTE REMOVAL

Line 183-188, 192-196: were the differences statistically significant? WE HAVE NOW INCLUDED THIS

Line 208-210: could higher N removal in deep wetlands compared to shallow wetlands be related to higher evaporation rate from shallow wetlands and therefore, N would get more concentrated within them? Did you measure in situ temperatures in wetlands? that could be higher in shallow wetlands. And in that case you would measure higher concentration in the outflow of shallow wetlands but effectively the removal efficiency per volume is higher than in deep wetlands? YES! BUT WE NOW INCLUDE TEMPERATURE IN METHODS TO CLARIFY THAT THE DIFFERENCE IS VERY SMALL (PROBABLY MAINLY BECAUSE RETENTION TIME WAS SHORTER FOR SHALLOW WETLANDS)

Line 219-222: do you have a reference for this? ADDED

Reviewer 3 Report

Please check my comments in the attached manuscript.

Author Response

Dear reviewer,

Thanks for valuable comments. Our response comes below, lines are according to the older pdf-version with your comments.

Line 63: Reformulated

Line 88: Not a new paragraph

Line 152: Information added

Line 161: Reformulated and references added

Line 179: The table shows p-values from ANOVA. Average values for N removal and removal coefficient are given in figures 2 and 3 with standard errrors

Line 239: References have been added to the discussion and a paragraph discussing practical implications has been added

Round 2

Reviewer 1 Report

I recommend the manuscript for publication in WATER